# Thermal effect on curved photovoltaic panels: Model validation and application in the Tabuk region

Nacer Badi [1,2,3]*, Azemtsop Manfo Theodore[4]*, Saleh A. Alghamdi[1,2,3], Ayshah S. Alatawi[1,5], Adnan Almasoudi[1], Abderrahim Lakhouit[6], Aashis S. Roy[7], Alex Ignatiev[8]

**1** Department of Physics, Faculty of Science, University of Tabuk, Tabuk, Saudi Arabia, **2** Nanotechnology Research Unit, University of Tabuk, Tabuk, Kingdom of Saudi Arabia, **3** Renewable Energy & Energy Efficiency Center, University of Tabuk, Tabuk, Saudi Arabia, **4** Center of Excellence on Solar Cells & Renewable Energy, School of Basic Science and Research, Sharda University, Greater Noida, India, **5** Sensor Networks and Cellular Systems Research Center, University of Tabuk, Tabuk, Saudi Arabia, **6** Department of Civil Engineering, Faculty of Engineering, University of Tabuk, Tabuk, Saudi Arabia, **7** Department of Chemistry, S.S.Tegnoor Degree College, Kalaburagi, Karnataka, India, **8** Department of Physics, University of Houston, Houston, TX, United States of America

* nbadi@ut.edu.sa (NB); azemsouleymane@yahoo.fr (AMT)

**Data Availability Statement:** There are legal restrictions placed on the data. The data underlying the results presented in the study are freely available upon request from the research sponsor,

## Abstract

This paper aimed to investigate the temperature effect on photovoltaic (PV) cell parameters. The PV cell parameters such as series and parallel resistances, diode ideality factor, and diode saturation current, are not considered in the reported stepwise modeling. The present work aims to improve available models used in the modeling and simulation of PV modules to support the researcher and power project developer. All the required temperature-dependent parameters are determined to model the simulated PV module with high accuracy using Simulink/MATLAB software. To validate the method, a 36-cell-50W solar panel with different radii of curvature is set up to assess solar power outputs under varying irradiance and temperature conditions. For the present application, the Tabuk region (Saudi Arabia) is chosen based on its location and climatic conditions. The method provided conformity to the measured power outputs for varying Global Horizontal Irradiance (GHI) and temperature conditions. The maximum power output of the PV module increases from 14.4 W to 25.8 W when the received solar power density varies from 307 W/m$^2$ to 526 W/m$^2$ depending on the level of curvature starting from a semi-cylindrical shape to a vaulted shape to a flat shape. The curved PV module shows slightly higher power variation with temperature as compared to the flat one. Above 25˚C, the power output is about 20% less at a maximum temperature of 65˚C. When the temperature drops below 25˚C, the power outputs increase about 6% and 11.5% for corresponding temperatures of 15˚C and 5˚C, respectively.

## 1. Introduction

Among renewable energies, photovoltaic generation is a particularly promising technology for electrical power generation owing to its eco-friendly and flexible operation. Many architects,

University of Tabuk Deanship of Scientific Research (srda@ut.edu.sa). The authors confirm they do not have special access privileges others would not have.

**Funding:** We would like to acknowledge the financial support towards this research from Deanship of Scientific Research (DSR), University of Tabuk, Tabuk, Saudi Arabia, under research Grant No. S-1441-0156.

**Competing interests:** The authors have declared that no competing interests exist.

designers, and manufacturers across the globe are investigating the usage of photovoltaics (PV) as a long-term energy source. Devices for solar charging on the go, solar lights [1, 2], solar plants [3–5], farming [6–8], buildings with solar power [9–12], and full integration within the textiles sector [13, 14] are applications of innovative PV materials design.

Various parameters, such as photovoltaic module temperature, installation angle, shade, irradiance, and orientation, must be addressed to improve PV module performance [15–20]. The temperature of the solar module and the irradiance should be considered the most important of these elements since they impact both the system's electrical efficiency and the energy performance of the operating solar energy harvester. While irradiance is directly proportional to the solar panel's electrical conversion efficiency, daily temperatures of about 60˚C cause a significant drop in photovoltaic performance and long-term damage.

Cylinders, domes, arches, and curved roofs are characteristics of rounded geometric shapes. Architectural elements that have curved surfaces are not compatible with flat solar panels [21–23]. The incidence of solar radiation reaching a PV surface and the curvature of the PV module determine the potential electrical power of the system. Most PV modules, whether crystalline silicon or thin films are produced as flat-plate modules. However, the increased availability of semi-flexible and flexible thin-film PV modules offers new possibilities, such as solar panels on irregularly curved surfaces. Thin, lightweight, durable, and flexible photovoltaic modules open large possibilities for solar harvesting power generation systems. To the best of our knowledge, studies of the performance of curved flexible solar panels are very limited in the literature [24–27].

To validate the method, a semi-cylindrical 36-cell- 50W solar panel is constructed and studied as an example for solar assessment of power output for varying irradiance and temperature conditions. The study was focused on the effect of temperature as a key parameter to be considered when sitting PV systems. The Tabuk region (Saudi Arabia) was selected based on its climatic conditions. It is considered a typical place in Saudi Arabia with a hot environment and where heat dissipation problems for PV panels are severe. The manuscript is organized as follows: Section 2 presents the measurement of solar radiation on a semi-cylindrical surface and the mathematical modeling of PV module. The results and discussions are presented in Section 3. Finally, Section 4 presents the conclusions of the manuscript.

## 2. Materials and methods

### 2.1. Measurement of solar radiation on a semi-cylindrical surface

The semi-cylindrical panel would occupy less footprint when compared to its planar counterpart having identical dimensions, same PV cell area, and working under the same conditions of irradiance and temperature. It can be expected to capture more sunlight in terms of direct and diffuse solar irradiance over an angular range of 180˚ when it is oriented in an east-west direction with its absorbing surface facing south and perpendicular to the sun's incoming rays. Because it is positioned on the sunny side, its surface does not have a shady part as in the case of a cylindrical shape. Although the panel would perceive an uneven solar irradiance at any climatic conditions, the curved surface shows a smoother spectrum of solar irradiance intensity and should not be treated as partial shading where several solar cells would distinctly receive less solar irradiance as compared to the rest of the cells. Furthermore, the uneven distribution of solar irradiance over time depends on the angle a PV cell is facing, not on nearby obstacles. When managing partial shading a few cells with low irradiance are excluded and treated separately in the net power assessment process. Also, for the power generation and optimization of the PV module as well as arrays placed on a curved surface such as the semi-cylindrical shape,

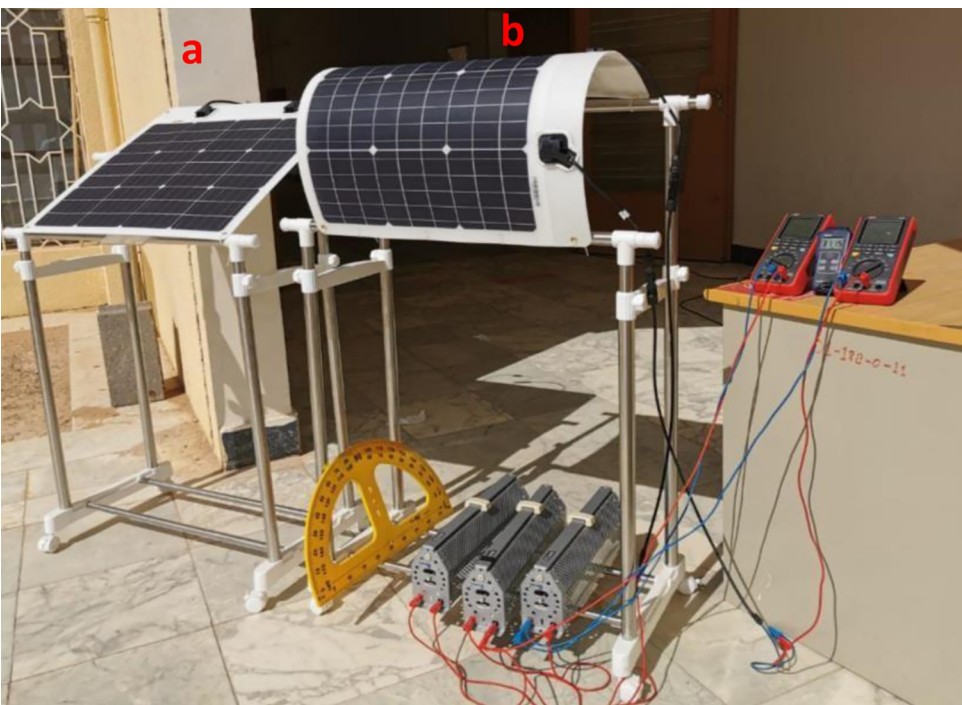

**Fig 1.** Snapshot of the experimental setup for MESM-50W solar panel: a) flat and b) semi-cylindrical.

this should be carried on with regards to the continuous spectrum of maximum power points (MPPs) of all the cells at the same time.

Measurements have been carried out on an identical flexible plane and semi-cylinder PV panels in full sunlight as shown in Fig 1. For the semi-cylindrical shape, we made sure the width or chord (w) of the circular arc is equal to twice the height (h) or sagitta by using a three-dimensional adjustable mechanical support allowing the side lengths of the semi-cylinder to be attached along the supporting bars in a quite close semi-cylindrical shape. The two identical PV panels were oriented in an east-west direction and positioned at 28.38 as the latitude of Tabuk city with their absorbing surfaces perpendicular to the sun's incoming rays. Of particular interest to PV installation is to consider the Global Horizontal Irradiance (GHI) which combines both Direct Normal Irradiance (DNI) and Diffuse Horizontal Irradiance (DHI) that hit at the same angle both flat and semi-cylinder photovoltaics panels on direction parallel to the central axis. GHI is the geometric sum of DNI and DHI components available on a flat surface. Therefore, GHI is expected to be reduced depending on the radius of the semi-cylindrical shape.

The percentage reduction needs to be evaluated in solar power density between the exposed semi-cylindrical surface and the tangential plane N–N depending on the inclination of the full sunlight as sketched in Fig 2.

The percentage reduction in power density is given by:

$$\Delta \in \% = \frac{\left[ {}^G\!/A_P - {}^G\!/_G \, / A_{cyl} \right]}{G/A_P} 100 \tag{1}$$

Where $A_{sc} = \pi . R . L$ is the semi-cylindrical area and $A_P = 2R, L$ is the tangential area. R and L are the radius and length of the semi-cylindrical shape, respectively. The ratio between $A_{sc}$

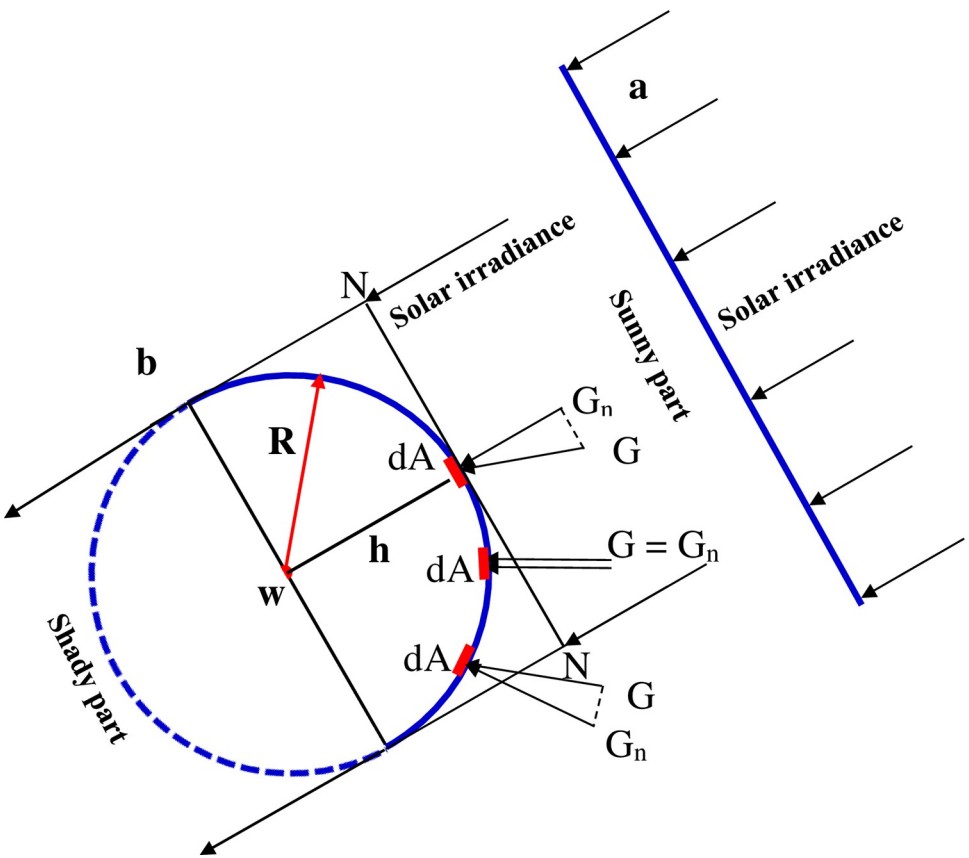

**Fig 2.** Tangential plane and semi-cylindrical surface exposed to full sunlight for a) Flat, b) curved PV panels. The dotted line represents the shady part of the full cylinder.

and $A_p$ is $\pi/2$. If the loss tangent component is ignored in all points of the semi-cylinder ($G = G_n$), the reduction in power density would be an upper limit of 36.3%. This value needs to be adjusted by measuring the maximum operating power outputs of the flat and semi-cylindrical PV modules with different radii of curvature under clear sky conditions. The output power is directly proportional to the incoming solar irradiance. This way the height of the cylindrical shape would not affect the measurement concerning the plane solar panel, as would be the case when using laboratory halogen lamps placed at a distance from the solar panel surface. In the latter case, the intensity of irradiance varies as we get closer to the source or further away from it and therefore affects the measurement of the output power.

## 2.2. Mathematical modeling of PV module

Available photovoltaic (PV) modeling procedures are based on presumptions and thus give uncertainties within the modeling data output. This is primarily due to some utilized parameters within the non-linear current-voltage characteristics of the solar-powered cell behavior, which if not determined, are taken as constants. Furthermore, the effect of temperature on some key parameters such as diode saturation current, diode ideality factor, and series resistance, have not been considered in the reported stepwise modeling. The present work aims to improve available models used in modeling and simulating PV modules to support the researcher and power project developer. All the needed temperature-dependent parameters were determined to precisely model the PV module.

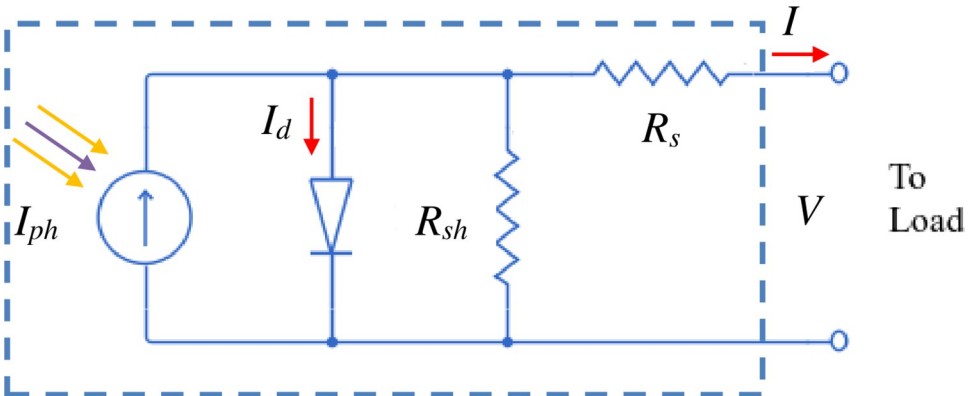

**Fig 3. An equivalent PV cells electrical circuit.**

Fig 3 depicts an equivalent PV cells circuit connected with series and shunt resistances $R_s$ and $R_{sh}$ [28]. By supposing these two internal values to be very small and very large, a simplistic model would ignore the values of the series and shunt resistances. The series resistance was included in a model of modest complexity, but the shunt resistance was regarded as quite substantial and hence was removed [29–33]. Because these resistances have an impact on the PV solar system's efficiency, they must be measured very precisely, as stated in the literature [34–39].

The generated current $I$ is given by the difference between the photocurrent ($I_{ph}$), diode current ($I_d$), and the shunt resistance current ($I_R sh$):

$$I = I_{ph} - I_d - I_{R_{sh}} \tag{2}$$

$$I = I_{ph} - I_o \left( e^{\frac{V+IR_s}{V_T}} - 1 \right) - \left( \frac{V + IR_s}{R_{sh}} \right) \tag{3}$$

$$V_T = \frac{N_s a K T_{opt}}{q} \tag{4}$$

where:

$I_o$: reverse saturation current

$q$: charge of the electron

$K$: Boltzmann constant

$V_T$: thermal voltage

$a$: solar cell ideality factor

$T$: ambient temperature

$N_s$: number of connected cells in series

Eq 2 represents the mathematical model that best matches experimental values. It is possible to accurately simulate the photovoltaic module by assessing the five parameters $I_{ph}$, $I_R sh$, $I_o$, $a$, and $R_s$. The data sheets of commercial PV modules do not provide these parameters. The following equation may be used to calculate $I_{ph}$.

$$I_{ph} = \left[ I_{sc} + K_{sc} \left( T - T_{ref} \right) \right] * G/1000 \tag{5}$$

where:

$I_{sc}$: short-circuit current

**Table 1. Electrical parameters of MESM-50W solar PV module.**

| | |
|---|---|
| Maximum power ($V_{max}$) | **50.02W** |
| Voltage at maximum power ($V_{mp}$) | 17.8 |
| Current at maximum power ($I_{mp}$) | 2.81 |
| Open-circuit voltage ($V_{oc}$) | 22.3 |
| Short-circuit current ($I_{sc}$) | 3.03 |
| Temperature coefficient of power | -0.41%/˚C |
| Temperature coefficient of voltage | -0.31%/˚C |
| Temperature coefficient of current | +0.05%/˚C |
| Efficiency of the cells (%) | 18.43% |
| Maximum operating voltage | 600VDC(IEC) |
| Tolerance in power output | ±3% |

**Note**: The electrical parameters are given under STC of irradiance of 1000W/m$^2$, spectrum of 1.5 air mass and cell temperature of 25˚C.

$K_{sc}$: temperature coefficient of $I_{sc}$

$G$: illumination of the PV module (W/m$^2$)

$T_{ref}$: reference temperature

Because $I_{ph}$, which is generated as a result of sunlight absorption, and solar irradiance are linearly related [40, 41], $G$/1000 is used to scale its value, which is the ratio of real illumination divided by reference illumination under standard test conditions (STC). The quantity $K_{sc}\Delta$T, which denotes the variation in photon current at a particular operating temperature, reflects the temperature effect on $I_{ph}$. So, under STC, this term is simply added to the current in a short circuit. As a result, Eq 5 yields the total photon current created for any solar irradiation and temperature circumstances.

As indicated in Fig 1, the current research is conducted by using a MESM-50W solar panel. Made in Germany with SunPower grade A monocrystalline solar cells, Me Solar flexible solar modules are excellent for a variety of surfaces and roofs. A total of 36 monocrystalline silicon solar cells make up the module linked in the series (3X12). The module dimensions are 650 x 505 x 2mm. The electrical parameters from the MESM-50W datasheet are listed in Table 1. This module has a strong copper foundation, which lowers the series resistance between cells and thus increases the efficiency of solar cells. Our analysis data were provided by a solar monitoring station located in Tabuk and which is based on one-minute measurements of Global Horizontal Irradiance (GHI) that hits a flat PV module. Table 2 shows the geographic position of Tabuk station, Tabuk, KSA.

The MESM-50W PV module's current versus voltage characteristic as provided by the manufacturer is shown in Fig 4, which was obtained experimentally using a curve tracer for I-V curves. The curve was created under STC test settings as shown in Table 1. The various interdependent parameters such as $I_{ph}$, $R_s$, and $I_Rsh$were determined following the same approach and methodology as recently reported by the authors [42].

**Table 2. Geographic position of Tabuk station, Tabuk, KSA.**

| | |
|---|---|
| Site name | **Tabuk University** |
| State | Tabuk |
| Latitude | 28.38287 |
| Longitude | 36.48396 |
| Elevation | 781m |

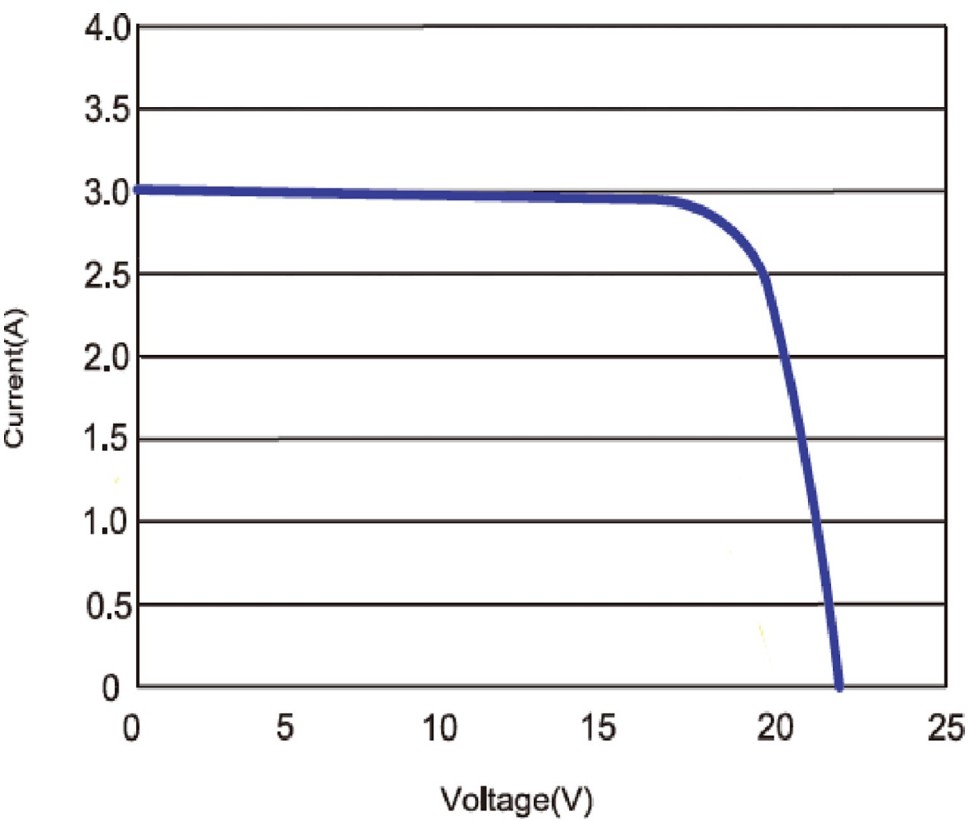

**Fig 4. I-V characteristic by varying irradiance for the MESM-50W PV module under STC conditions.**

## 3. Results and discussions

The variables $I_o$, $a$, and $R_s$ were obtained by starting with some initial or guess values then the system of coupled nonlinear is solved simultaneously to get to the correct values of the variables [42].

Fig 5 shows the interdependent variables $I_o$, $a$, and $R_s$ models which were simultaneously extracted versus temperatures up to 65˚C. The best fit models of these variables are also presented. The exponential data models were excellent fits for both $I_o$ and $a$ parameters, except for the portion between 5˚C and 15˚C for the parameter $a$. The polynomial fit worked also well for $R_s$ under temperatures above 15˚C. Overall, $I_o$ and $a$ increases with temperature while $R_s$ decreases in the temperature range from 15˚C to 65˚C. We currently don't have a realistic explanation as to why this anomaly is appearing at temperatures below 15˚C. The obtained fitting models are just models which are well suited for this given experimental situation at lower thermal stress, but never perfectly. We believe that such parameters' behavior might just result from the temperature dependence of another quantity, having a different physical origin than the true $a$ and $R_s$.

Table 3 summarizes the models fitting equations and parameters including standard data errors for the variables $I_o$, $a$, and $R_s$. The best fit for temperature-dependent parameters to the models fitting equations, including standard data errors, is presented in the range of 15˚C to 65˚C.

Table 4 gives the MPPs of the MESM-50W module at different curvatures as defined by the radius of curvature starting from semi-cylindrical (R = 22.5 cm) to flat shape. The measurements were taken at the Renewable Energy Laboratory under solar irradiance of 1450 W/m$^2$,

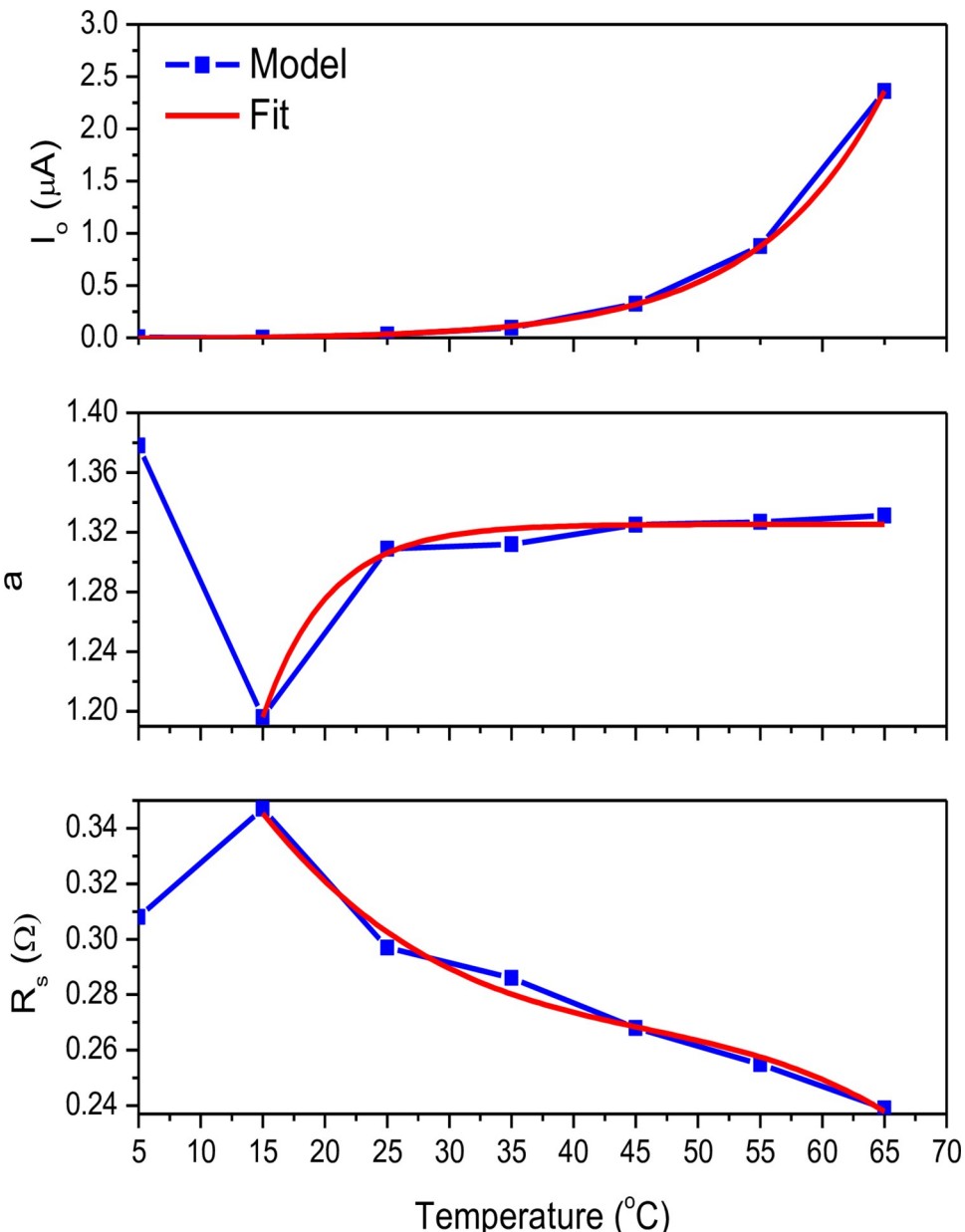

**Fig 5. Extracted parameters $I_o$, $a$, $R_s$ and best fitting models versus temperature.**

at an ambient temperature of 35˚C, and at 7% humidity. The aim was to evaluate the percentage reduction $\Delta\in \%$ in power density when using curved surface as compared to a flat PV panel with identical dimensions. This reduction depends on the geometry setting and not on the amount of solar irradiance. The upper limit of 36.3% in power density reduction for the semi-cylindrical shape (diameter = 45 cm), as shown in Section 2, is only credible if we ignore the loss tangent component in all points of the semi-cylinder ($G = G_n$). The upper limit of loss is found above to be equal to 36.3% for the semi-cylindrical shape (diameter = 50 cm). The measurement gives $\Delta\in \% = 48.7$. The upper limit of Eq 2 is adjusted by considering the loss tangent component of the direct radiation to the semi-cylinder, which is equal to 12.4%.

**Table 3. Models fitting equations and parameters including standard data errors.**

| | Equations | Parameters | Standard Errors |
|---|---|---|---|
| $I_o$ | $y = Ae^{\left(-\frac{x}{b}\right)} + y_o$ | $A = 3.961.10^{-9}$ | $3.089.10^{-10}$ |
| | | $y_o = -1.248.10^{-8}$ | $6.043.10^{-9}$ |
| | | $b = -10.163$ | $0.123$ |
| $a$ | $y = Ae^{(R_o x)} + y_o$ | $A = -2.216$ | $1.392$ |
| | | $y_o = 1.325$ | $0.004$ |
| | | $R_o = -0.190$ | $0.042$ |
| $R_s$ | $y = B_1 x + B_2 x^2 + B_3 x^3 + y_o$ | $B_1 = -0.011$ | $0.00338$ |
| | | $B_2 = 2.223$ | $9.227.10^{-5}$ |
| | | $B_3 = -1.611.10^{-6}$ | $7.643.10^{-7}$ |
| | | $y_o = 0.469$ | $0.037$ |

## 3.1. Simulink-MATLAB simulation

The MESM-50W PV module was modeled using a Simulink-MATLAB application. Fig 6 depicts the finished model that gathers solar energy, converts it into electrical power, and outputs reactions typical for a particular solar irradiance in W/m² along with cell temperature in ˚K. Regarding the Simulink-MATLAB environment, the model is built on the usage of prior expressions in mathematics indicating the net current output of the PV module. The required model is equipped with subsystems that are developed and connected in order to implement PV current, shunt current, and photocurrent output [42]. A resistive load drives the output voltage, allowing to measure and monitor the voltage and current output.

Fig 7 shows the Simulink/Simscape model for a continuous spectrum of MPPs determination of the PV module under varying irradiance and temperature. The MPP value is well known to increase linearly with the short circuit current, which in turn, increases linearly with irradiance. The model calls for the "Powergui" block as a graphical user interface that displays the maximum values of power and voltage in a continuous mode. The attached load was adjusted to 6.2 Ω to adapt the 50 Watts PV panel. The MPPs variation with temperature along with its best linear fit model is presented in Fig 8. The parameters of the fitting model are included in the inset table.

Overall, the MPPs drop linearly as the temperature rises. When the temperature rises over STC and the sun illumination remains constant, the power output drops by roughly 14.5 percent when the temperature hits 65˚C. No issue is foreseen while the cell is operating at maximum power because of the high quality of the MESM-50W solar panel. The estimated power temperature coefficient is around -0.39 percent /˚C, which is quite close to the value given in Table 1 by the manufacturer. When the temperature falls below STC, however, the power output increases by around 7.4%, exceeding the maximum power of the rated PV panel. The most important implication is the maximum number of cells that may be connected in series without exceeding the panel's maximum voltage rating. As a result, running the solar panel in cold locations is a problem since $V_{oc}$ grows as the temperature drops. This might be a problem in terms of the overall number of PV panels that can be linked in series.

**Table 4. Gives the MPPs of the MESM-50W module at different curvatures.**

| R (cm) | ∞ | 53.5 | 36.8 | 29.5 | 25.4 | 22.5 |
|---|---|---|---|---|---|---|
| MPPs (Watts) | 38.5 | 35.15 | 31.4 | 28.15 | 26.2 | 22.5 |
| Δ∈ % | 0 | 8.7 | 18.4 | 26.9 | 31.9 | 41.5 |

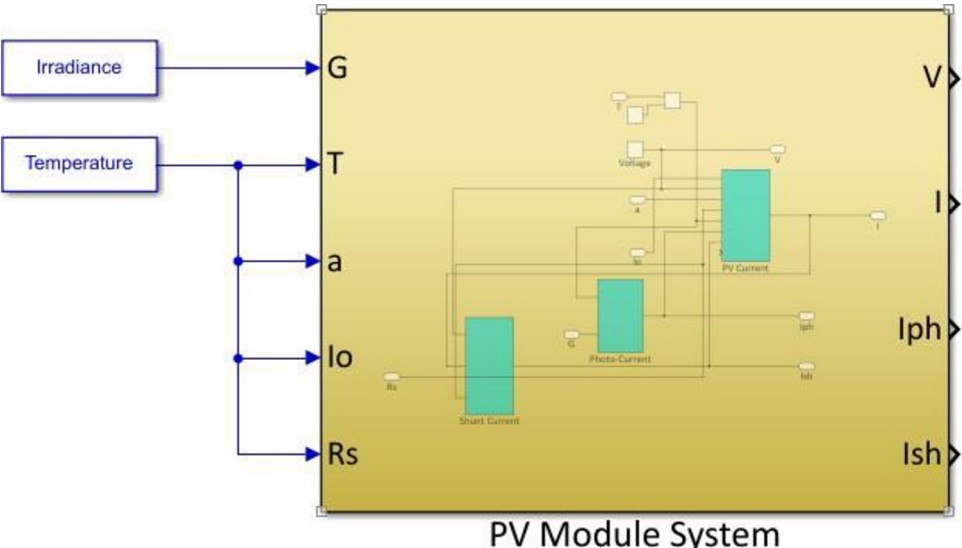

**Fig 6. Complete block diagram of the simulated MESM-50W PV module.**

## 3.2. Effect of irradiance and temperature on power output

The performance of the solar PV module has been determined in Simulink/MATLAB by using the average daily total GHI for the solar irradiance components over the 1-year study period from December 2015 until November 2016 [43]. The average daily total GHI for Tabuk is 6314.7 Wh/m$^2$ for a maximum daytime duration of 12 hours, which converts to an average irradiance of about 526 W/m$^2$ with an average daily total temperature of 25°C. The solar irradiation G and the temperature values T at the site are two factors that have a significant impact on the photovoltaic panel's response in terms of voltage, output current, and power. The semi-cylindrical P-V curves shown in Fig 9 correspond to varied irradiance intensity levels depending on the amount of curvature. As expected, the flat PV module received higher solar irradiance and therefore generates more power output. The maximum power output of the PV module increases from 14.4 W to 25.8 W when the received solar power density varies from 307 W/m$^2$ to 526 W/m$^2$ depending on the level of curvature starting from a semi-cylindrical shape to a vaulted shape to a flat shape. Therefore, solar power reduction depends on the exposed (projected) area and not on the climatic condition—amount of solar irradiance and ambient temperature. As the measurements gave a percentage reduction ranging from 48.7% (semi-cylindrical shape) to 8.7% (for stretched vaulted shape), Fig 9 shows the corresponding radii values of the projected areas (PA) which clearly exhibit the relationship between the electrical power output and the received solar power, which then translates into solar efficiency of the PV panel.

The temperature also significantly affects the power output characteristics. To determine the P-V response accurately, the temperature-dependent $I_o$, $a$, and $R_s$ parameter values are used. Fig 10 illustrates the P-V curves of the semi-cylindrical module for various temperatures ranging from 5 to 65 degrees Celsius at a 526 W/m$^2$ irradiance. The open-circuit voltage and the maximum power point are the two conditions where the temperature has the greatest influence. Because the voltage temperature coefficient is -0.31%/°C, $V_{oc}$ decreases as the temperature rises. The current temperature coefficient represents a very small shift of +0.05%/°C. The decrease in $V_{oc}$ as a function of temperature is mostly due to an increase in dark saturation current $I_o$ (saturation current at open-circuit voltage, $I_{ph}$ = 0) as a function of temperature.

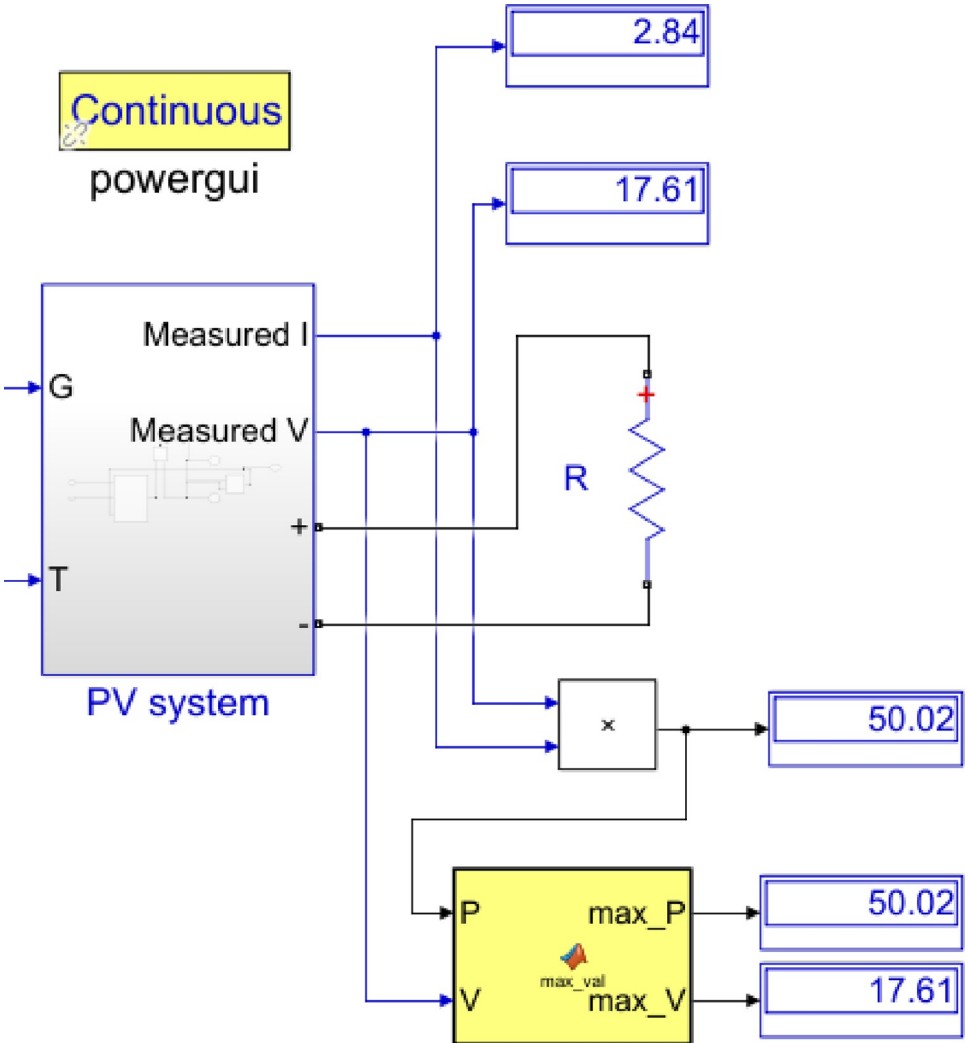

**Fig 7. Simulink/Simscape model for continuous spectrum of MPPs of the MESM-50W module.**

Because there are more electron-holes created at higher temperatures, the band gap of the solar cell material slightly decreases with temperature for a given irradiance, resulting in a very small increase in generated current. The increase in the ideality factor *a* with temperature as shown in Fig 5 would substantially *increase* the dark saturation current so that a solar cell with a high ideality factor would typically have a lower turn on voltage. The semi-cylindrical module shows less power variation with temperature as compared to the flat one. Above 25°C, the power output is about 20% less as the temperature reaches a value of 65°C. However, as the temperature is below 25°C, the power output goes up by about 6% and 11.5% for temperatures of 15°C and 5°C, respectively.

## 4. Conclusions

This paper aimed to investigate the temperature effect on photovoltaic (PV) cell parameters. The PV cell parameters such as series and parallel resistances, diode ideality factor, and diode saturation current, are not considered in the reported stepwise modeling. Available models used in the modeling and simulating of PV modules were improved to support the researcher

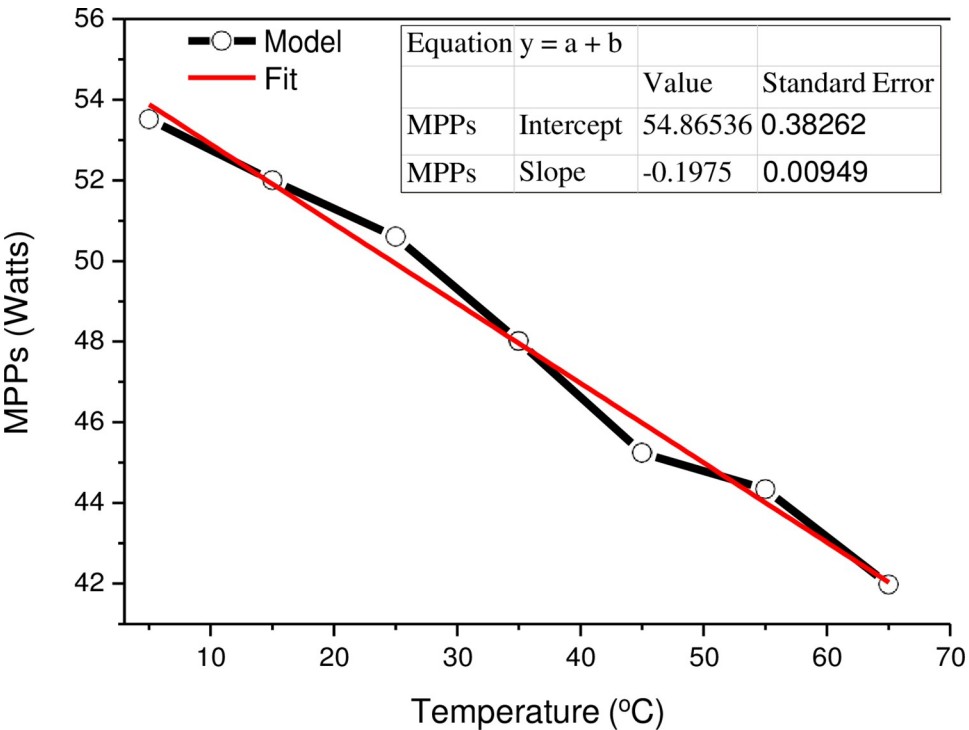

**Fig 8. MPPs and best fit models with varying temperature.**

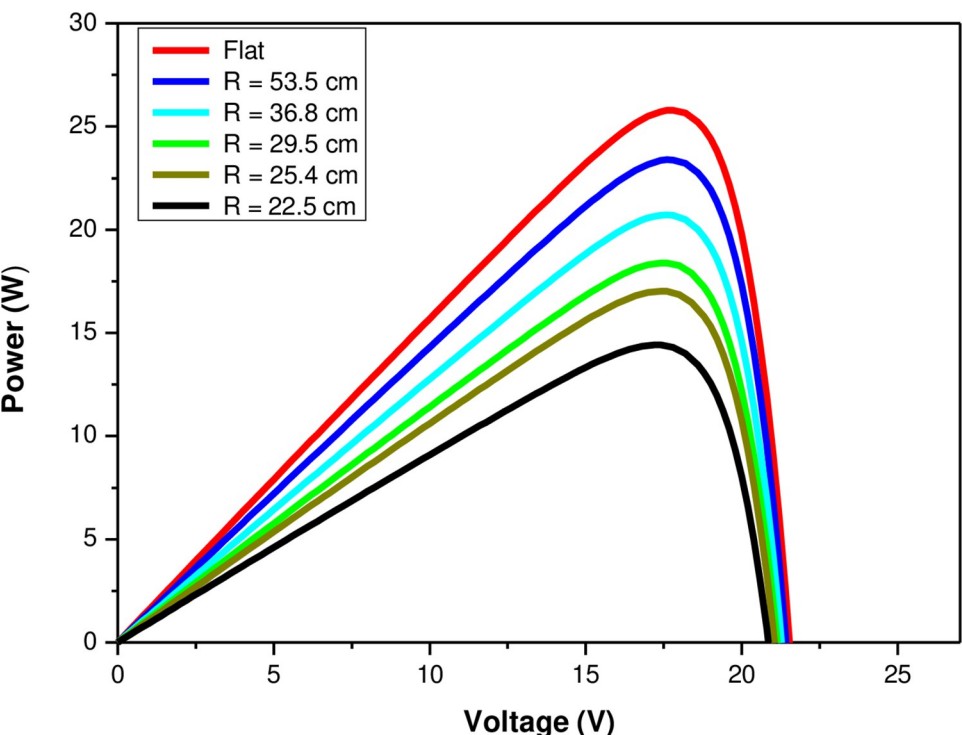

**Fig 9. Power output characteristics with varying the radius of curvature.**

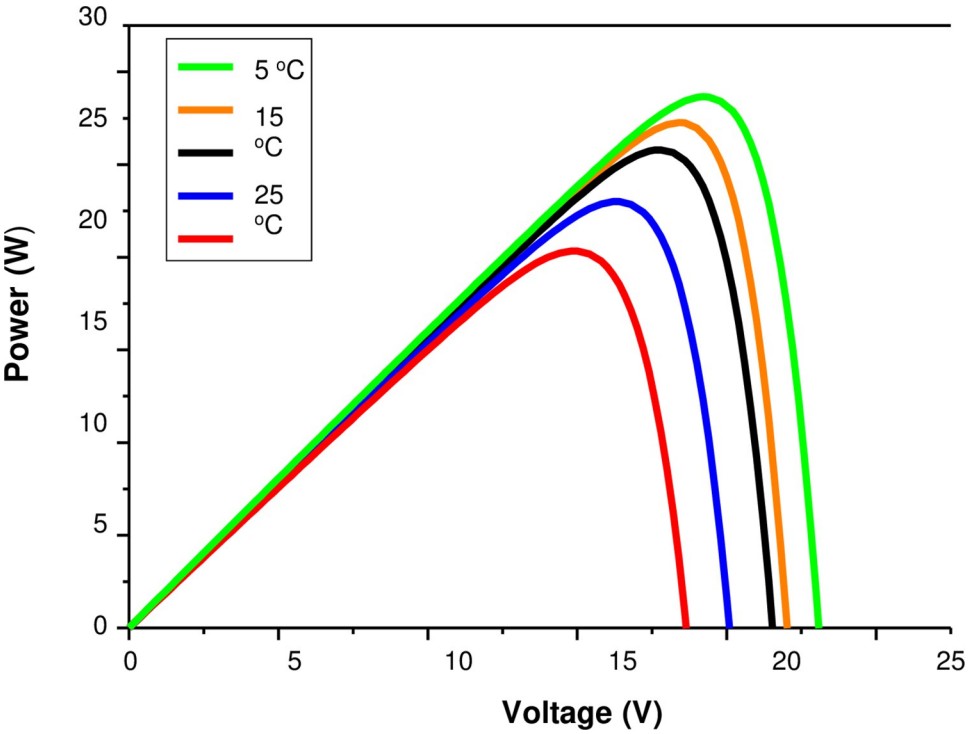

**Fig 10. P-V characteristics output for various operating temperature values.**

and power project developer. The needed temperature-dependent parameters were determined and modeled with high accuracy in a simulated PV module using Simulink/MATLAB software. The method was validated by taking experimental measurements on both flat and curved 36-cell-50W solar panels for solar assessment of power outputs under varying irradiance and temperature conditions. The effect of temperature as a key parameter was to be considered when siting PV systems. For a direct application, the Tabuk region (Saudi Arabia) was selected based on its location and climatic conditions. The method provided conformity to the measured power outputs for varying Global Horizontal Irradiance (GHI) and temperature conditions. The maximum power output of the PV module increases from 14.4 W to 25.8 W when the received solar power density varies from 307 W/m$^2$ to 526 W/m$^2$ depending on the level of curvature starting from a semi-cylindrical shape to a vaulted shape to a flat shape. Therefore, solar power reduction depends on the exposed (projected) area and not on the climatic condition—amount of solar irradiance and ambient temperature. The measurements gave a percentage reduction in solar power ranging from 48.7% for semi-cylindrical shapes to 8.7% for stretched vaulted shapes. The curved PV module shows slightly higher power variation with temperature as compared to the flat one. Above 25°C, the power output is about 20% less at a maximum temperature of 65°C. When the temperature drops below 25°C, the power outputs increased about 6% and 11.5% for corresponding temperatures of 15°C and 5°C, respectively.

## Author Contributions

**Investigation:** Alex Ignatiev.

**Methodology:** Aashis S. Roy.

**Resources:** Adnan Almasoudi.

**Visualization:** Ayshah S. Alatawi, Abderrahim Lakhouit.

**Writing – original draft:** Nacer Badi.

**Writing – review & editing:** Azemtsop Manfo Theodore, Saleh A. Alghamdi.

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
