## [Decision Letter · Decision Letter 0]

8 Aug 2022

PONE-D-22-16013Thermal effect on curved photovoltaic panels: model validation and application in Tabuk regionPLOS ONE

Dear Dr. Badi,

Thank you for submitting your manuscript to PLOS ONE. After careful consideration, we feel that it has merit but does not fully meet PLOS ONE’s publication criteria as it currently stands. Therefore, we invite you to submit a revised version of the manuscript that addresses the points raised during the review process. Editor's and reviewer's comments are appended below.

We look forward to receiving your revised manuscript.

Kind regards,

Mahendra Singh Dhaka, Ph.D.

Academic Editor

PLOS ONE

Journal Requirements:

“We would like to acknowledge the financial support towards this research from Deanship of Scientific Research (DSR), University of Tabuk, Tabuk, Saudi Arabia, under research Grant No. S-1441-0156.”

“We would like to acknowledge the financial support towards this research from Deanship of Scientific Research (DSR), University of Tabuk, Tabuk, Saudi Arabia, under research Grant No. S-1441-0156.

The funders had no role in study design, data collection and analysis, decision to publish, or preparation of the manuscript.”Please include your amended statements within your cover letter; we will change the online submission form on your behalf.

Additional Editor Comments :

In addition to the reviewer’s comments, the authors need to revise the manuscript as under and by providing a list of five potential reviewers those may review this manuscript:

1. The language should be polished thoroughly.

2. The scientific interpretations of the facts should be strengthened throughout.

3. Recent relevant references should be incorporated, if possible, also PONE transections.

4. The conclusion part should be strong based upon the findings.

Reviewers' comments:

Reviewer's Responses to Questions

**Comments to the Author**

1. Is the manuscript technically sound, and do the data support the conclusions?

Reviewer #1: Partly

2. Has the statistical analysis been performed appropriately and rigorously? 

Reviewer #1: Yes

3. Have the authors made all data underlying the findings in their manuscript fully available?

Reviewer #1: No

4. Is the manuscript presented in an intelligible fashion and written in standard English?

Reviewer #1: No

5. Review Comments to the Author

Reviewer #1: PLOS ONE

Thermal effect on curved photovoltaic panels: model validation and application in Tabuk region

Manuscript Number: PONE-D-22-16013

General: I, we, our, etc. should not be used.

Abstract: English editing is required.

To validate the method, we have set up and studied a 36-cell-50W solar panel with different radii of curvature is set up to assess solar power outputs under varying irradiance and temperature conditions.

For present application, the Tabuk region (Saudi Arabia) is chosen based on its location and climatic conditions.

What are the climatic conditions made the basis for site selection and why not Dhahran or any other location?

Message is not meaningful: Depending on the radius of curvature, an increase from 14.4 W to 25.8 W in the PV module power output was measured in the range of 307 W/m2 to 526 W/m2 of absorbed solar radiation.

Mixed tenses: In abstract, past and present tenses have been used. Please make uniform.

Introduction: English editing is required.

Many architects, designers, and manufacturers across the globe are investigating the usage of photovoltaics (PV) as a long-term energy source. in their products.

The temperature of the solar module and the irradiance should be considered the most important—Did authors considered these two factors or it is a statement from literature?

In Section 2, first paragraph, the statements are conflicting or confusing. Are the authors explaining advantages of curves surfaces or disadvantages compared to other geometries?

The panels were oriented in an east-west direction and positioned at 28.38 as the latitude of Tabuk city. Shouldn’t it be south facing????

Is it geographic or Demographic position of Tabuk station???

How did the authors obtain Figure 4?

Please clarify the sentence: Figure 5 shows the interdependent variables , , and models which were simultaneously extracted versus temperature up to 65oC.

Figure 5: Authors show the modelled values and the curve fitting? Where is the measured data?

Same is the case in Figure 8.

Figure 9 and 10: These are the simulated values. Where are the measured values at different radius of curvatures?

Discussion section should be re-written for clarity and also English grammar should be rectified.

Following papers are suggested for enhancing the background of the topic of the manusvript:

1. Parametric Analysis and Optimization of a Novel Photovoltaic Trombe Wall System with Venetian Blinds: Experimental and Computational Study, Case Studies in Thermal Engineering, 34, 101958, March 2022.

2. A Comprehensive Global Review of Building Integrated Photovoltaic Systems, FME Transactions, March 2021, 49(2), 253-268.

3. Prospects of large-scale photovoltaic-based power plants in the Kingdom of Saudi Arabia, Engineering Reports, Article No. 12398, 1-16.

4. Global solar radiation and energy yield estimation from photovoltaic power plants for small load, International Journal of Green Energy, 14(4), 490-498. (JIF = 1.171). http://dx.doi.org/10.1080/15435075.2016.1278374

5. Performance evaluation of a PV (photovoltaic) module by back surface water-cooling for hot climatic conditions, Energy – The International Journal 59, 445-453.

6. Performance Evaluation of an off-grid Photovoltaic System in Saudi Arabia, Energy – The International Journal, v. 46(1), 451-458.

7. Usage of Photovoltaics in Automated Irrigation System, Renewable Energy, 23(1), (2001), pp. 17-26.

8. Feasibility study of the grid connected 10MW installed capacity PV power plants in Saudi Arabia. Renewable and Sustainable Energy Reviews, 80, 319 – 329.

9. Performance Evaluation of a PV Module Under Climatic Conditions of Dhahran, Saudi Arabia, Energy Exploration & Exploitation 33(6), 909-930.

6. PLOS authors have the option to publish the peer review history of their article (what does this mean?). If published, this will include your full peer review and any attached files.

Reviewer #1: **Yes: **SHAFIQUR REHMAN A A

---

## [Author Response · Author response to Decision Letter 0]

26 Aug 2022

Response to Reviewers is enclosed.

---

## [Decision Letter · Decision Letter 1]

19 Sep 2022

Thermal effect on curved photovoltaic panels: model validation and application in the Tabuk region

PONE-D-22-16013R1

Dear Dr. Badi,

We’re pleased to inform you that your manuscript has been judged scientifically suitable for publication and will be formally accepted for publication once it meets all outstanding technical requirements.

Kind regards,

Mahendra Singh Dhaka, Ph.D.

Academic Editor

PLOS ONE

Additional Editor Comments (optional):

Reviewers' comments:

Reviewer's Responses to Questions

**Comments to the Author**

1. If the authors have adequately addressed your comments raised in a previous round of review and you feel that this manuscript is now acceptable for publication, you may indicate that here to bypass the “Comments to the Author” section, enter your conflict of interest statement in the “Confidential to Editor” section, and submit your "Accept" recommendation.

Reviewer #1: All comments have been addressed

2. Is the manuscript technically sound, and do the data support the conclusions?

Reviewer #1: (No Response)

3. Has the statistical analysis been performed appropriately and rigorously? 

Reviewer #1: N/A

4. Have the authors made all data underlying the findings in their manuscript fully available?

Reviewer #1: Yes

5. Is the manuscript presented in an intelligible fashion and written in standard English?

Reviewer #1: Yes

6. Review Comments to the Author

Reviewer #1: No comments, as the authors have incorporated almost all the comments in their revised version of the manuscript.

7. PLOS authors have the option to publish the peer review history of their article (what does this mean?). If published, this will include your full peer review and any attached files.

Reviewer #1: **Yes: **Shafiqur Rehman

---

## [Editor Report · Acceptance letter]

7 Oct 2022

PONE-D-22-16013R1 

Thermal effect on curved photovoltaic panels: model validation and application in the Tabuk region 

Dear Dr. Badi:

I'm pleased to inform you that your manuscript has been deemed suitable for publication in PLOS ONE. Congratulations! Your manuscript is now with our production department. 

Kind regards, 

on behalf of

Dr. Mahendra Singh Dhaka 

Academic Editor

PLOS ONE